# Effect of preconditioning on propofol-induced neurotoxicity during the developmental period

**Satoshi Shibuta** [1,2,3]*, **Tomotaka Morita**[2], **Jun Kosaka**[2]

**1** Research Institute, Nozaki Tokushukai Hospital, Daito-city, Osaka, Japan, **2** Department of Anatomy, School of Medicine, International University of Health and Welfare, Narita, Chiba, Japan, **3** Department of Anesthesiology, Kyoaikai Tokushukai Hospital, Hakodate, Hokkaido, Japan

* satoshishibuta66@gmail.com

## Abstract

At therapeutic concentrations, propofol (PPF), an anesthetic agent, significantly elevates intracellular calcium concentration ($[Ca^{2+}]i$) and induces neural death during the developmental period. Preconditioning enables specialized tissues to tolerate major insults better compared with tissues that have already been exposed to sublethal insults. Here, we investigated whether the neurotoxicity induced by clinical concentrations of PPF could be alleviated by prior exposure to sublethal amounts of PPF. Cortical neurons from embryonic day (E) 17 Wistar rat fetuses were cultured *in vitro*, and on day *in vitro* (DIV) 2, the cells were preconditioned by exposure to PPF (PPF-PC) at either 100 nM or 1 μM for 24 h. For morphological observations, cells were exposed to clinical concentrations of PPF (10 μM or 100 μM) for 24 h and the survival ratio (SR) was calculated. Calcium imaging revealed significant PPF-induced $[Ca^{2+}]i$ elevation in cells on DIV 4 regardless of PPF-PC. Additionally, PPF-PC did not alleviate neural cell death induced by PPF under any condition. Our findings indicate that PPF-PC does not alleviate PPF-induced neurotoxicity during the developmental period.

## Introduction

Propofol (PPF), whose chemical structure is shown in Fig 1, is one of the most extensively used intravenous anesthetic agents in clinical settings, especially in pediatric and obstetric patients, because of its rapid onset and reversibility. However, PPF-associated neurotoxicity at the developmental stages remains a concern [1]. Moreover, because PPF crosses the placenta, caution should be exercised when administering it to pregnant women.

PPF-induced neurotoxicity in immature neurons has been previously illustrated [2–9], where, in one of the reports, we have demonstrated that at clinically relevant concentrations, PPF induced a significant increase in intracellular calcium ($[Ca^{2+}]i$) and neural cell death during the developmental period in rats. Recently, several animal studies have attempted to attenuate PPF-induced neurotoxicity [10–12].

Preconditioning (PC) is defined as a treatment that elicits changes at the biomolecular level that enable specialized tissues to better tolerate a serious adverse events by exposure to

**Data Availability Statement:** All relevant data are within the paper and its Supporting Information files.

**Funding:** YES: Shibuta received following fund: This work was supported by JSPS (Japan Society

for the Promotion of Science) KAKENHI Grant Number 19K09359. https://nrid.nii.ac.jp/ja/nrid/1000020324767/ Funders did not play any role in the study design, data collection and analysis, decision to publish, or preparation of the manuscript.

**Competing interests:** The authors have declared that no competing interests exist.

**Fig 1. Chemical structure of propofol (PPF).**

sublethal insults [13, 14]. Exposure to low concentrations of toxic chemicals that are toxic at moderate concentrations can lead to neuroprotection against a major lethal insult [14]. For instance, in combination with rat primary cultured neurons, low dose N-methyl-D-aspartate (NMDA) PC induced neuroprotection against glutamate cytotoxicity [15, 16].

The purpose of this study is to determine whether the neurotoxicity induced by clinical concentrations of PPF during the developmental period is alleviated by prior exposure to sublethal amount of PPF.

## Materials and methods

### Ethical approval

In the present study, we purchased and used 12 pregnant rats in total. All animals were treated in strict accordance with the National Institutions of Health and International University of Health and Welfare (IUHW) guidelines for the care and treatment of laboratory animals. The study protocol was approved by the Animal Care Committee of the IUHW (reference number 17025). All efforts were made to minimize the number of animals used and their suffering.

### Chemical reagents

Chemical reagents used in the present experiment were purchased from the following sources. PPF, 5-fluoro-2'-deoxyuridine (5-FU), poly-l-lysine, streptomycin, penicillin, and Pluronic acid F-127 were obtained from Sigma–Aldrich (St. Louis, MO, USA). Dulbecco's modified Eagle's medium (DMEM), dimethyl sulfoxide (DMSO), and KCl were obtained from Wako Pure Chemical Industries, Ltd. (Osaka, Japan). Supplement Minus Antioxidants (AO) and horse serum (HS) were obtained from Gibco BRL (Carlsbad, CA, USA) and from Dako (Carpinteria, CA, USA), respectively. Trypsin was purchased from Difco Lab (Detroit, MI, USA), fetal calf serum (FCS) from ICN Biochemicals (Costa Mesa, CA, USA), and B-27 Fluo4-AM from Dojindo (Kumamoto, Japan). PPF concentrations in the present study are comparable to the moderate concentrations prescribed for clinical use [9, 17].

### Cell culture and preconditioning

Primary cortical cultured neurons from rats were prepared as previously described [18, 19]. All Wistar rats used were purchased from Nihon SLC (Hamamatsu, Japan). Rat fetuses were

removed from sevoflurane-anesthetized pregnant Wistar rats on embryonic day 17 (E17). Fetal rat brains were carefully removed using a microscope. The meninges and blood vessels were removed and titrated with a Pasteur pipette several times, then cerebral cortical neurons were treated with 0.25% trypsin in phosphate buffered saline (PBS) at 37˚C for 20 min. Dispersed cells were diluted to a concentration of 0.6–1.0 × 10$^6$ cells/ml in DMEM. The medium contained 4% HS, 8% FCS, 2% B-27 Supplement Minus AO, 50 IU/ ml penicillin, and 50 μg/ ml streptomycin. The suspension was placed in a poly l-lysine–coated film-bottom dish with a diameter of 35 mm (FD10300; Matsunami Glass Ltd, Osaka, Japan) for calcium imaging or in 2 mm-grid tissue culture dishes (Nunc, Naperville, IL, USA) for cytotoxicity measurements. Grid tissue culture dishes were used to observe the same neurons, as previously described [20, 21].

On day *in vitro* (DIV) 2, cells were exposed to the vehicle, DMSO+ PBS, alone or PC by exposing the cells to PPF (PPF-PC) at 100 nM or 1 μM concentration diluted DMSO+PBS, which did not affect the survival rate of the neurons in our preliminary experiments (SR: 0.9956 ± 0.006; 0,9961 ± 0.012; both of them showed P> 0.9)

Twenty-four hours after PC (DIV 3), the cultured medium was completely replaced. Cultured cells were treated with 5-FU (5 μg/ml) to prevent the proliferation of non-neuronal cells. Thereafter, cultured neurons were maintained in DMEM with 8% FCS, 4% HS, and 2% B-27 supplement in incubators under the following conditions: 5% $CO_2$, 100% humidity, and 37˚C temperature.

## Calcium imaging experiments

Calcium imaging experiments were conducted as previously described [9, 22–24]. On DIV 4, the intracellular calcium concentration ($[Ca^{2+}]i$) was measured using a fluorescence measurement system (Aquacosmos®; Hamamatsu Photonics, Hamamatsu, Japan) and an inverted phase contrast microscope (Axiovert 200®; Carl Zeiss, Oberkochen, Germany). Neurons were transferred to a normal bath solution (NBS; 137 mM NaCl, 5 mM KCl, 2.5 mM $CaCl_2$, 1 mM $MgCl_2$, 10 mM HEPES, pH 7.3, and 22 mM glucose) containing a $Ca^{2+}$-sensitive indicator, 10 μM Fluo4-AM, and 0.025% Pluronic acid F-127 for 30 min at 23–25 degrees Celsius.

The cultured neurons were rinsed twice with fresh NBS before being placed on a microscope stage. A 150 W xenon lamp was used for fluorescence excitation (450–490 nm) of neurons. Fluorescence images (16 bit, 512 × 512 pixels; 2 × 2 binned) were collected using a cooled charge-coupled device camera (Orca ER®; Hamamatsu Photonics) linked to the microscope, with an oil-immersion objective lens (Fluor® ×40, oil, numerical aperture 1.30; Carl Zeiss). A series of images was collected with an integration time of 2 s for 120 s. The excitation light was blocked using a filter exchanger (C8214; Hamamatsu Photonics) to avoid potential cell damage during exposure. Processing software (Aquacosmos®, Hamamatsu Photonics) was used for image analysis.

An increase in $[Ca^{2+}]i$ indicates an enhancement in fluorescence intensity [25]. Data are expressed as the mean relative fluorescence within a defined region of each cell. To estimate the effect of PPF on primary cortical cultured neurons, the mean maximum change in fluorescence intensity (Fmax) upon addition of PPF was measured and normalized to the baseline fluorescence acquired prior to PPF application (F0) as illustrated in Fig 2.

## PPF (or vehicle) application

PPF solutions and vehicles were prepared shortly before use. KCl solutions were also prepared for neurons in case they did not respond to the anesthetic. As previously described, if a neuron

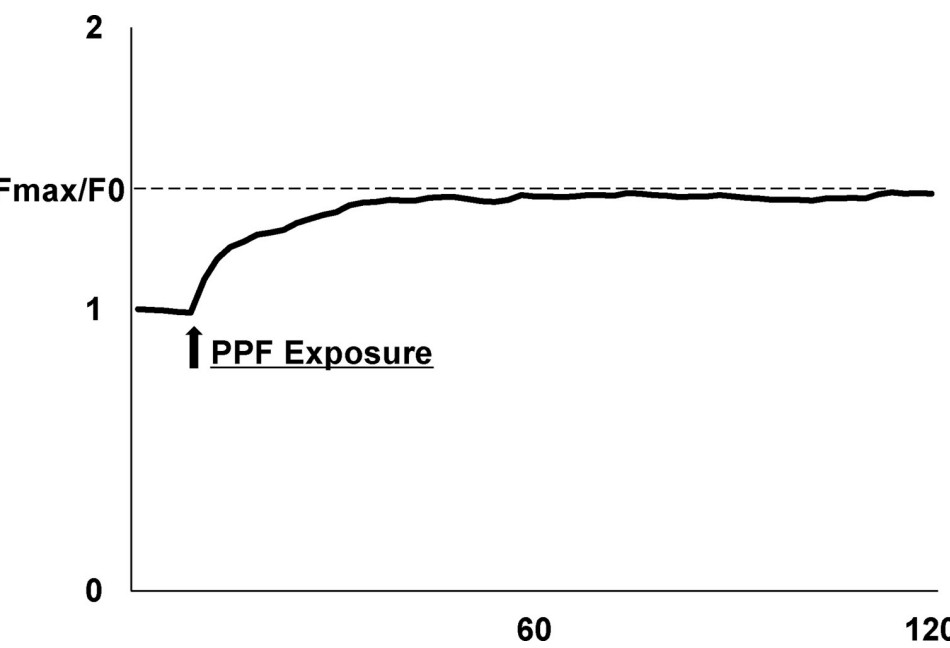

**Fig 2. The intensity of the fluorescence (F/F0) over time in a cultured primary neuron.** PPF application led to an increase in the temporal intracellular calcium concentration ([Ca$^{2+}$]i). The arrow indicates PPF exposure and onset. To evaluate the PPF application effect, we observed Fmax/F0 as height calculated from the [Ca2 +]i response in the neuron.

did not react to KCl treatment, it was considered dead, and its data was excluded from the analysis [23, 24].

PPF solution was administered by a second observer blinded to the study design and treatment protocols on the areas near the observed neurons, but not directly onto the cells, using a P-200 Gilson Pipetman with a maximum effective volume of 50 μl (final volume of 1 ml solution: 10 or 100 μM PPF) as reported previously [9, 23].

We evaluated the responses of neurons in 46 cultured dishes. Fifty neurons per dish were randomly selected for [Ca2 +]i. To assess the effect of PPF application, we observed the mean value of Fmax/F0 as height calculated using the calcium response curves of cells.

Among the 50 neurons, the number of neurons with an Fmax larger than 1.5 were counted. The Fmax values of the 50 neurons that responded to calcium were also calculated. These procedures were performed by a second researcher blinded to the study design and treatment protocol. We investigated the height of the cells exposed to vehicle (DMSO in NBS) as a control application and calculated the height ratio (HR) as follows:

The height of the calcium response to PPF application in the cultured neurons / the height of the calcium response in the control culture (PPF-PC = 0 on DIV 2). Hence, the HR of control cells was defined as 1.

## PPF-elicited neurotoxicity

As previously reported [26–28], neurotoxicity was assessed according to Shibuta's model, using a photosystem (Axiovert 25, Carl Zeiss; Nikon D90, Nikon, Japan). Primary cultured cortical neurons were exposed to PPF for 24 h. The survival rate was evaluated at the end of each procedure.

There were 9 groups and 69 cultured dishes in this study. We assessed the survival rates of the neurons exposed to the vehicle (DMSO in PBS) as a control.

Three photomicrographs were taken shortly before PPF or vehicle exposure (DIV 3) as well as at the end of the experiment (DIV 4). An observer was able to determine the specific location of each culture dish because of the grid arrangement. At the end of the experiments, cells were exposed to 0.4% trypan blue dissolved in PBS to stain non-viable cells, and photomicrographs were taken again in the same area as before the PPF exposure. The viable neurons remained unstained, whereas non-viable cells were either stained with trypan blue or washed away from the culture dish. Therefore, viable neurons were easily distinguished from dead neurons. Approximately 200–300 neurons were manually counted and examined per culture dish.

A second observer, blinded to the arrangement of the photographs, study design, and treatment protocol, repeated all manual counts to ensure count accuracy. At the end of the experiment, survival rates were calculated as follows:

The number of unstained cells / the total number of cells shortly before the experiment.

Survival ratio (SR) was calculated as follows:

Survival rate of the given dish / the survival rate of the control dish (PPF-PC = 0 on DIV 2 and PPF = 0 on DIV 3).

Therefore, the SR of control neurons was defined as 1.

## Statistical analysis

We calculated HR and SR to minimize the effect of various conditions associated with primary culture, including temperature, vehicle, and the concentration of sevoflurane used after the removal of fetuses. HR and SR factors were used for the comparison of groups as these conditions were the same in the sister-control culture dishes [9, 23, 24].

Statistical comparisons for the number of cells whose Fmax exceeded 1.5, HR, and SR were conducted using JMP Pro® 13.2.0 software (SAS Insti. Inc., Cary, North Carolina). The data are expressed as the mean and the standard error of the mean (SEM). The differences between the means were assessed using ANOVA (analysis of variance), followed by the two-tailed Tukey–Kramer honestly significant difference test as a post-hoc test. Statistical significance was set at $P < 0.05$.

## Results

### PPF-evoked $[Ca^{2+}]i$ rise on DIV4

First, we indicated the number of neurons with a height (Fmax/F0) > 1.5 per 50 neurons; 1.80 ± 0.66 of 50 neurons without PPF-PC on DIV 2 had a height exceeding 1.5 with vehicle on DIV 4. PPF at either 10 or 100 μM significantly increased the number of neurons with a height > 1.5 (7.0 ± 2.02 and 19.6 ± 2.71, respectively) compared to vehicle (Table 1 and Fig 3).

As for neurons with PPF-PC at 100 nM on DIV 2, 0.75 ± 0.48 of 50 neurons exceeded 1.5 in vehicle. PPF at either 10 or 100 μM elicited a significant increase in the number of neurons for which the height was > 1.5 (10.4 ± 2.62 and 22.3 ±3.09, respectively) compared to vehicle. Meanwhile, for neurons with PPF-PC at 1 μM on DIV 2, and with vehicle (DMSO in NBS) on

**Table 1. The number of neurons that had FMax > 1.5 in 50 cells.**

| | | PPF-PC | | |
|---|---|---|---|---|
| **PPF** | | **0** | **100 nM** | **1 μM** |
| | **0** | 1.80 ± 0.66 | 0.75 ± 0.48 | 0.00 ± 0.00 |
| | **10 μM** | 7.00 ± 2.02 | 10.40 ± 2.62 | 4.00 ± 1.22 |
| | **100 μM** | 19.60 ± 2.71 | 22.33 ± 3.09 | 22.50 ± 4.78 |

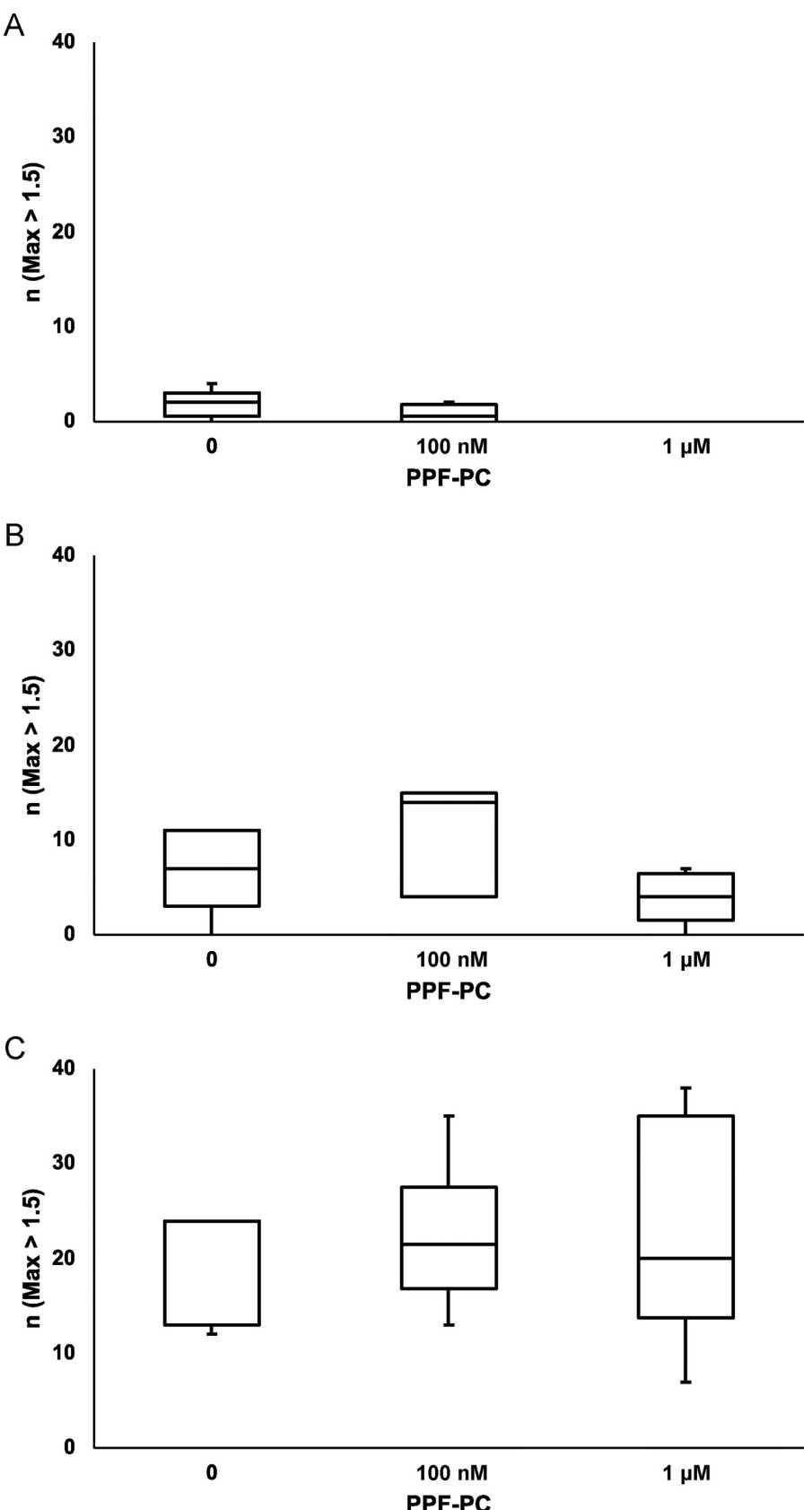

**Fig 3. The number of the neurons that had Fmax (i.e., height) > 1.5 in 50 neurons.** When neurons were exposed to vehicle (DMSO dissolved in NBS) on DIV 4, almost no neurons showed a significant height increase (exceeding F max > 1.5), regardless of exposure to PPF-PC on DIV 2 (A). Both 10 μM (B) and 100 μM (C) PPF significantly increased the number of neurons whose Fmax was > 1.5 compared to vehicle. However, PPF-PC did not affect the number of neurons whose F max was > 1.5. The differences between the means were calculated using analysis of variance (ANOVA), followed by the Tukey–Kramer honestly significant difference test as a post-hoc test.

DIV 4, no neurons exceeded Fmax = 1.5 height. PPF at either 10 or 100 μM induced a significant increase in the number of neurons for which the height was > 1.5 (4.0 ± 1.22 and 22.5 ± 4.78, respectively) compared to vehicle.

Next, the HRs of neurons exposed to 10 or 100 μM PPF on DIV 4 (Table 2 and Fig 4) were calculated. The presence of PPF-PC on DIV 2 did not significantly influence HR.

Fig 5 shows the time-course images of neurons from all experimental groups, demonstrating the $Ca^{2+}$ fluorescence reacting to PPF exposure. To determine whether this PPF-evoked rise was caused by the influx of extracellular $Ca^{2+}$, calcium ion free-NBS solutions (139.5 mM NaCl, 22 mM glucose, 10 mM HEPES, 5 mM KCl, 1 mM $MgCl_2$, and 1 mM EGTA) was used. Under these conditions, no PPF-elicited $[Ca^{2+}]i$ increase was observed (none of the neurons exceeded Fmax > 1.1); hence, we concluded that the observed $[Ca^{2+}]i$ increase was attributed to the influx of $Ca^{2+}$ from outside the neurons.

## PPF-PC did not affect PPF elicited neuronal death on DIV3-4

Fig 6 shows the micro-images of primary cultured neurons with transmitted light with or without PPF-PC exposure on DIV 3 and 4 (24 h following PPF or vehicle exposure). Exposure to the vehicle (DMSO in PBS) did not induce neural cell death in any treatment group; conversely, PPF exposure (10 or 100 μM) on DIV 3 significantly decreased SR, as shown in Table 3 and Fig 7. This PPF-induced neurotoxicity was not influenced by the presence of PPF-PC.

## Discussion

In this study, we performed both morphological and calcium-imaging evaluations to acquire insights into the effect of PC against PPF-induced neurotoxicity. Further, we assessed whether PPF-PC could alter calcium dynamics or improve cell resistance to PPF exposure during the early neural development period.

PPF is one of the most frequently used intravenous anesthetic agents used for the induction and maintenance of general anesthesia and sedation. By potentiating the inhibitory neurotransmitter gamma aminobutyric acid (GABA), PPF exerts a hypnotic effect. Although PPF is expected to be a neuroprotective agent due to its antioxidant [29] or free radical scavenger properties [30], it is a neurotoxin during the early developmental period [7, 8, 31]. This raises serious concerns regarding PPF use in neonates, infants, children, and adolescents [32]. However, major epidemiologic studies, such as the PANDA and GAS studies [33, 34], have not demonstrated substantial results associated with anesthetic-elicited neurotoxicity. Since PPF is

**Table 2. Height Ratios (HR) of neurons exposed to PPF (N) = number of dishes.**

| PPF | | PPF-PC | | |
|---|---|---|---|---|
| | | **0** | **100 nM** | **1 μM** |
| | **0** | 1 (5) | 0.95 ± 0.01 (4) | 0.95 ± 0.02 (5) |
| | **10 μM** | 1 (5) | 1.06 ± 0.03 (5) | 0.99 ± 0.05 (5) |
| | **100 μM** | 1 (5) | 1.01 ± 0.03 (6) | 1.01 ± 0.04 (6) |

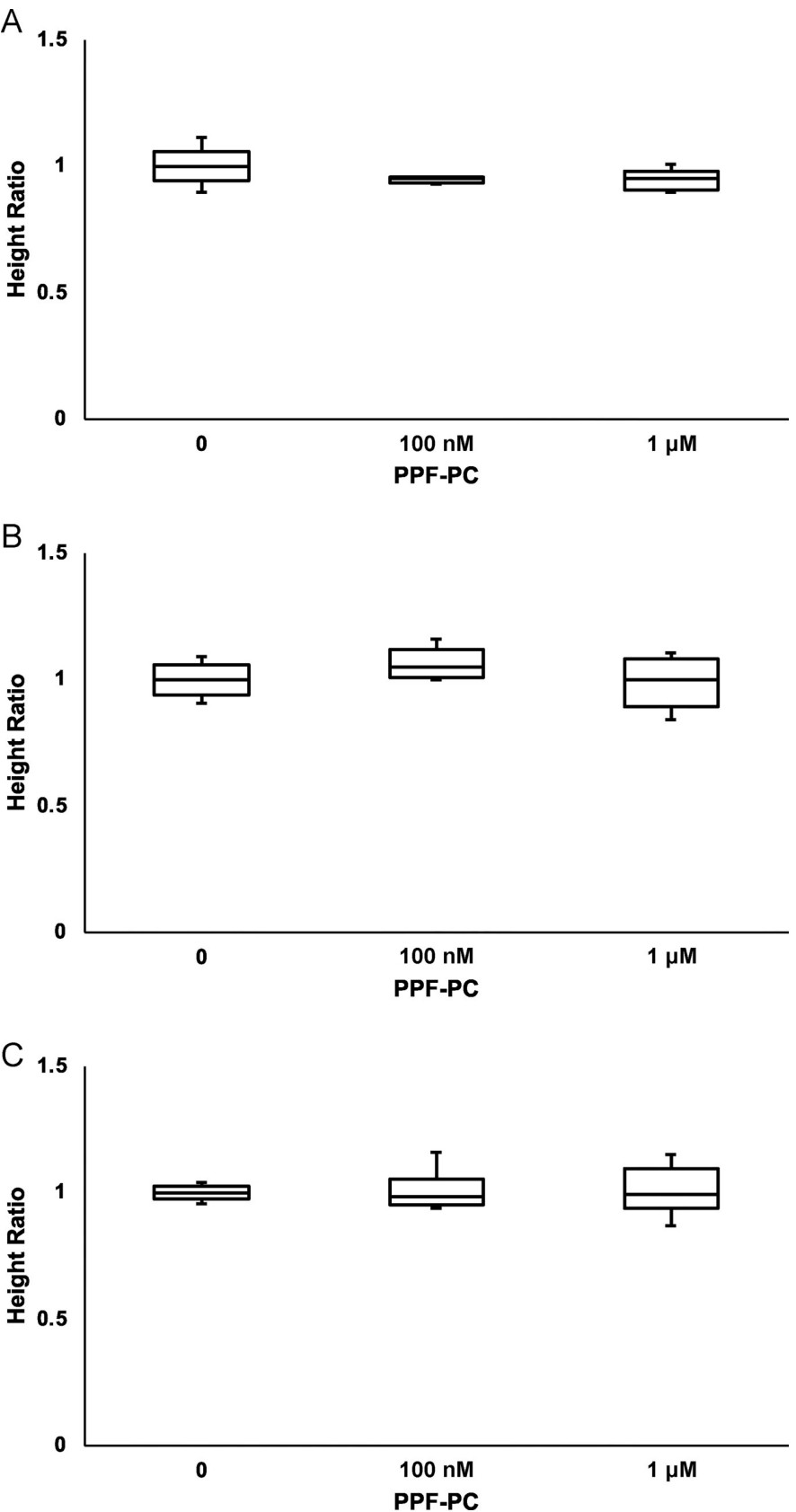

**Fig 4. Height ratios (HR) of neurons in response to vehicle or PPF.** HR of all groups were irrelevant to PPF-PC on DIV 2; (A) vehicle (DMSO dissolved in NBS). PPF exposure at 10 μM (B) and 100 (C) μM increased Fmax/F0, significantly. PPF-PC on DIV 2 did not affect Fmax/F0. The differences between the means were calculated using analysis of variance (ANOVA), followed by the Tukey–Kramer honestly significant difference test as a post-hoc test.

a GABA receptor agonist, the relevance of GABA shift and PPF-elicited neurotoxic effects in immature neurons have been investigated [35]. During the developmental period, GABA functions as an excitatory neurotransmitter; hence, it is likely that PPF induces neurotoxic effects via excitatory mechanisms [35].

Although the precise mechanisms underlying PPF-induced neurotoxicity remain nebulous, some studies have suggested hypotheses including dysregulation of cellular calcium levels, transcription factor dysfunctions, mitochondrial dysfunction, alternation of receptors, inflammation, and alternation of actin dynamics [2–9, 36]. Calcium is associated with various cellular processes including protein synthesis, cell differentiation and proliferation, and gene expression. Therefore, calcium homeostasis is important, especially during the developmental period, in which neurons have an insufficient capacity to regulate calcium homeostasis, and excessive $[Ca^{2+}]i$ rise elicits mitochondrial calcium overload. Therefore, developing neurons are critically sensitive to agents or factors that change their intracellular calcium concentration. Consequently, this will induce ROS production and neurotoxicity due to the disruption of the mitochondrial respiratory chain subsequent to the failure of ATP production [37–43].

We have previously [9] reported that PPF evoked a marked $[Ca^{2+}]i$ increase and caused death of neurons on DIV 4 and 8, but not 13 at therapeutic concentrations in primary cortical cultured neurons obtained from E17 Wistar rats. These results are compatible with the GABA shift and a critical window of the developmental period for the rodent brain [44–46], in which the brain networks are actively being organized; therefore, any pathological alterations can negatively affect these networks.

In our present study, we used DIV2-4 neurons to investigate PPF-PC against PPF-induced neurotoxicity since neural death and calcium dysregulations were the hallmarks of this stage in our previous research [9]. This was consistent with the GABA-shift period, which is around 7 POD in rodents, equivalent to the last trimester of pregnancy in humans. The GABA-shift period corresponds to the critical stages for neuronal growth. Consequences of depolarization with GABA can be observed within the initial 7 POD in rats, and similarly in the third trimester of pregnancy to the first six months after birth in human [3, 47–49].

Although previous human epidemiologic studies have not reported meaningful data regarding PPF-elicited neuronal toxic effects, it remains necessary to identify some measures to alleviate PPF-induced neurotoxicity during the developmental period considering data on PPF-elicited neurotoxicity from animal studies. Previous studies have shown that xenon and hypoxic PC attenuated PPF-elicited neurotoxicity [10–12].

PC has been observed in various organs, including the brain [14]. Additionally, using rat primary cultured neurons, PC with a low dose of NMDA elicited neuroprotection against glutamate insults [15]. Although the precise mechanism underlying PC is not fully understood, several mechanisms such as, receptor activation, intracellular signaling cascades, mitochondrial structural changes, alternations in the activation status and/or level of expression of proteins, and modulation of calcium homeostasis, have been implicated [13, 15, 16, 50–52].

In the present study, neither calcium-imaging nor morphological experiments showed any significant positive effects of PPF-PC. The concentrations of PPF used in the current study were moderate (10 μM) and high (100 μM) for clinical use and toxic to DIV 3 and 4 neurons [9, 17]. Meanwhile, PPF-PC, at either 100 nM or 1 μM, was not toxic even in DIV 3 and 4 neurons. Therefore, it is reasonable to assess whether the PC effects by exposure to low doses of

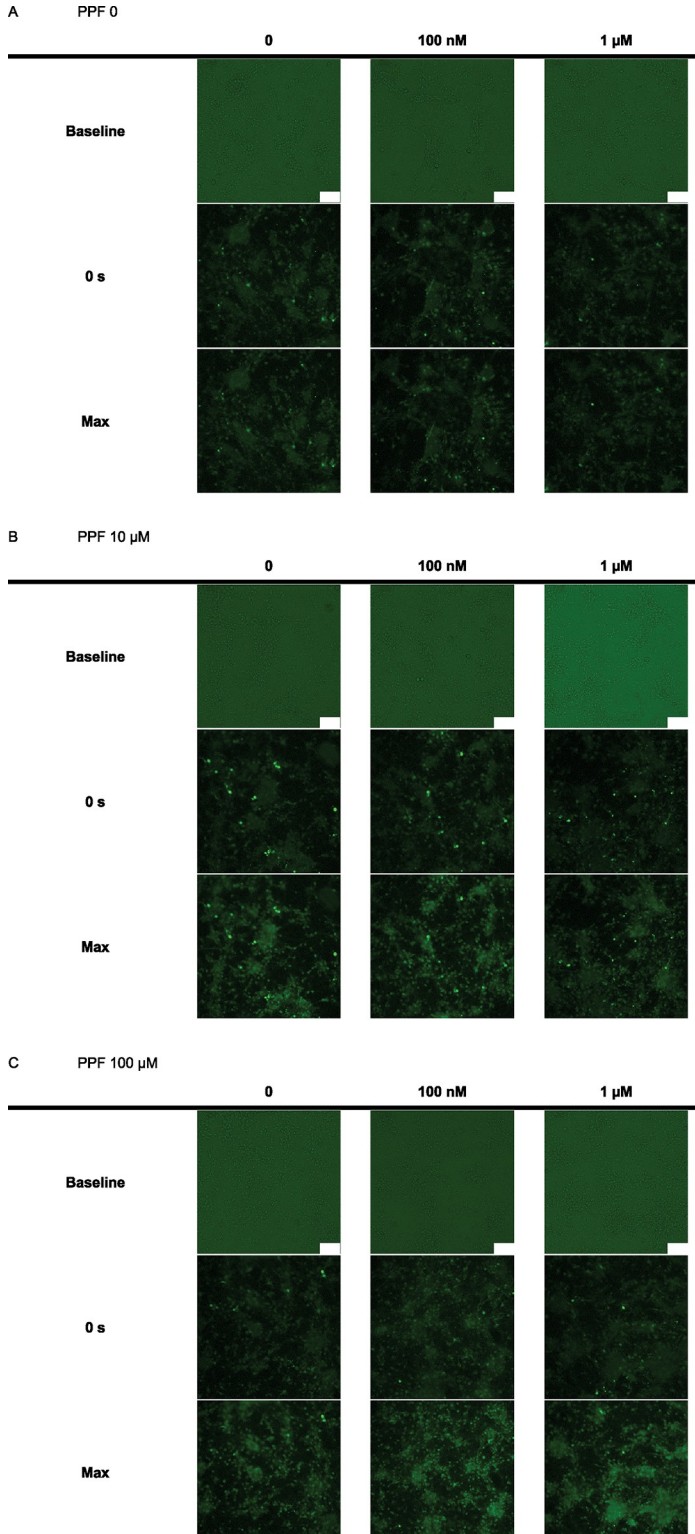

**Fig 5. Images of the time course of neurons from all groups demonstrating the changes of [Ca2 +]i in response to PPF application on DIV 4.** In the vehicle groups (dimethyl sulfoxide [DMSO]), the fluorescence intensities of the neurons showed little change (A), whereas in all 10 μM (B) and 100 μM (C) PPF exposure groups, the fluorescence intensities of neurons on DIV 4 were significantly high, and were irrelevant to PPF-PC on DIV 2. 0s: shortly before DMSO of PPF application. Max; most nearly the moment when Fmax is recorded. Since each neuron showed its own Fmax (peak moment of intracellular calcium concentration), not all the neurons showed Fmax at the same time in these images. Scale Bar = 100 μm.

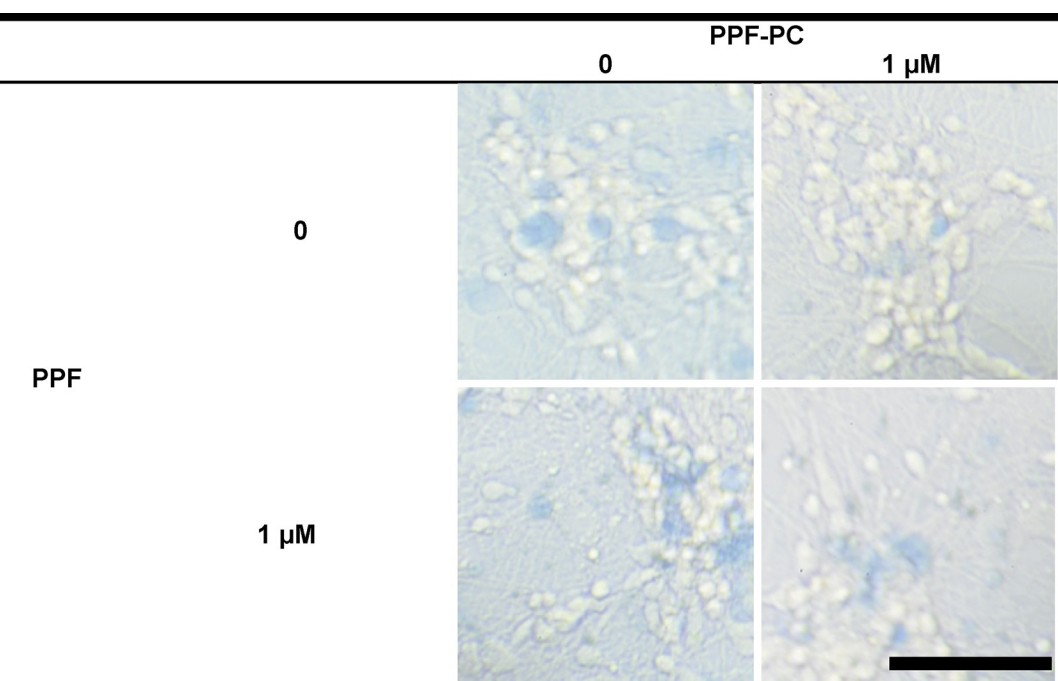

**Fig 6. Transmitted light microphotographs of primary cultured cortical neurons exposed to vehicle (DMSO in PBS) or 100 μM PPF on DIV 3, taken 24 h after the exposure.** Regardless of PPF-PC on DIV 2, PPF exposure on DIV 3 significantly induced neuronal death compared to vehicle. Scale Bar = 100 μm.

PPF (100 nM or 1 μM) might alleviate a major adverse event, exposure to moderate and clinically used concentrations (10 or 100 μM) of PPF. In addition, PPF-PC exposure was set for 24 h, and since longer exposure was not possible in the present study, there is little prospect that PPF-PC is neuroprotective.

This study has some limitations. First, we could not completely adapt the window of vulnerability and PPF exposure periods (POD 1–14, E 19–21, respectively). We used DIV 2–4 neurons critically injured using the same clinical concentrations of PPF as those in our previous study [9]. Therefore, it is appropriate to use cortical cultured neurons of this period despite not completely matching the window of vulnerability. Second, the number of glial cells, which support the survival and maintenance of neurons, was relatively small in our culture dishes. Therefore, the results obtained using the present *in vitro* model might not be completely equivalent to the results obtained using *in vivo* models.

## Conclusions

Our findings suggest that PPF-PC does not attenuate intracellular calcium elevation or neural death elicited by a clinical dose of PPF *in vitro*. A limited prospect suggests that PPF-PC protects neurons from PPF-induced neurotoxicity during the neural developmental period. Our

**Table 3. Survival ratios (SR) of neurons exposed to PPF (N) = number of dishes.**

| PPF | | PPF-PC | | |
|---|---|---|---|---|
| | | 0 | 100 nM | 1 μM |
| | 0 | 1 (7) | 0.98 ± 0.06 (7) | 0.99 ± 0.04 (7) |
| | 10 μM | 0.84 ± 0.06 (7) | 0.85 ± 0.04 (7) | 0.84 ± 0.02 (7) |
| | 100 μM | 0.82 ± 0.03* (8) | 0.82 ± 0.04* (9) | 0.80 ± 0.02*(10) |

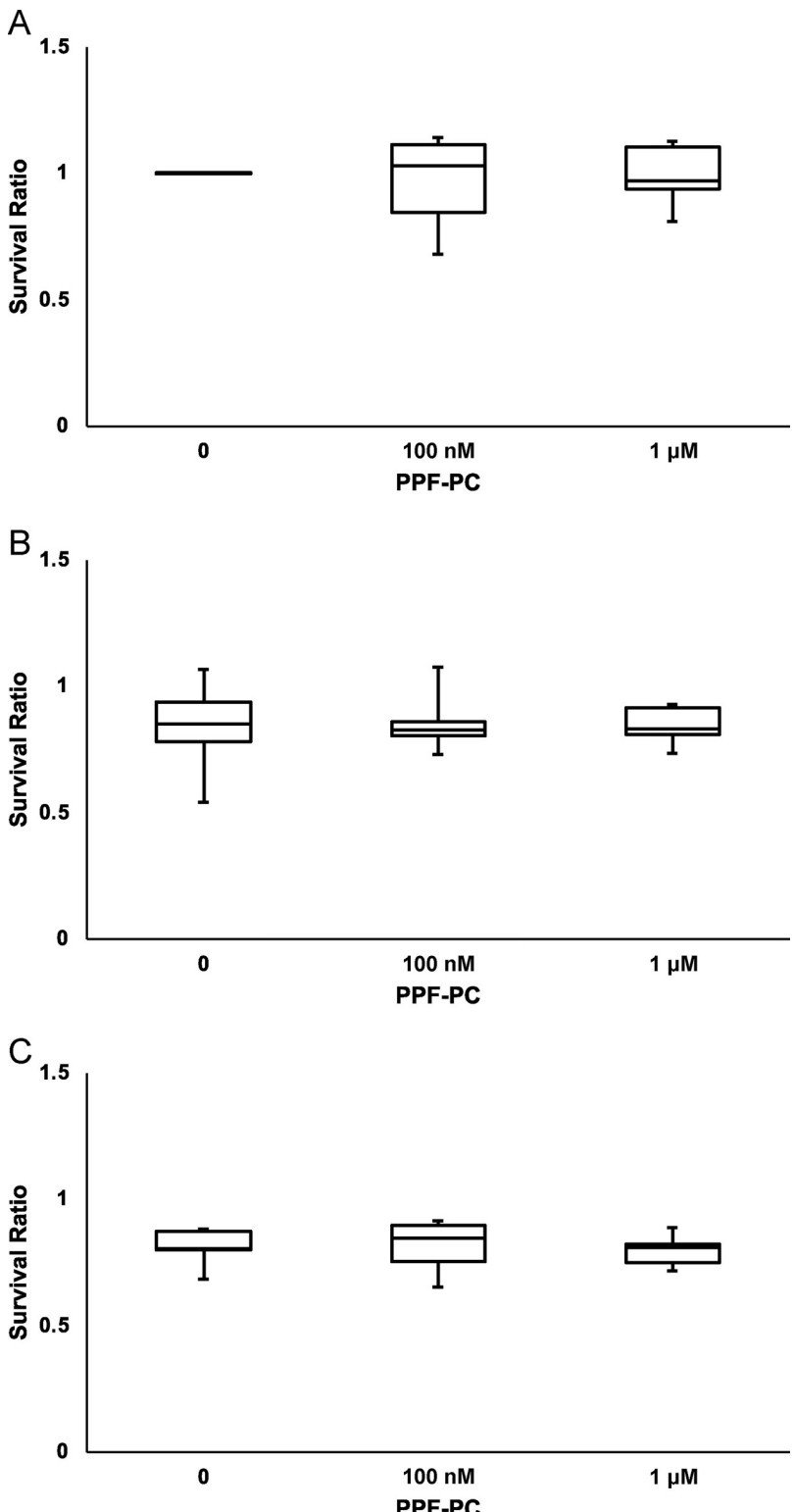

**Fig 7.** Survival ratio (SR) of primary cultured cortical neurons exposed to vehicle (DMSO in PBS (A); PPF at (B) 10 or (C) 100 μM) for 24 h during DIV 3 and 4. Although exposure to PPF elicited a significant neuronal death, the decrease in SRs on DIV 3 and 4 were irrelevant to PPF-PC on DIV 2. The differences between the means were calculated using ANOVA, followed by the Tukey–Kramer honestly significant difference test as a post-hoc test.

results provide new insights into the safety of PPF in pediatrics, suggesting that the use of PPF during the developmental period should be restricted whenever necessary.

## Supporting information

**S1 Checklist. ARRIVE checklist.**
(PDF)

**S1 File. PC PPF imaging PO.** Data of Ca-imaging Experiments.
(XLSX)

**S2 File. PPF PC SR PO.** Data of morphological Experiments.
(XLSX)

**S3 File. PPF100.** Data of Ca-imaging Experiments using calcium ion free-NBS solutions.
(XLSX)

## Acknowledgments

The authors thank the members of the Department of Anesthesiology, Osaka University, for their assistance in this project.

## Author Contributions

**Conceptualization:** Satoshi Shibuta.

**Data curation:** Tomotaka Morita.

**Investigation:** Satoshi Shibuta, Tomotaka Morita, Jun Kosaka.

**Methodology:** Satoshi Shibuta.

**Software:** Tomotaka Morita.

**Visualization:** Tomotaka Morita.

**Writing – original draft:** Satoshi Shibuta.

**Writing – review & editing:** Jun Kosaka.

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
