## [Decision Letter · Decision Letter 0]

15 Jun 2022

PONE-D-22-08060Effect of preconditioning on propofol-induced neurotoxicity during developmental periodPLOS ONE

Dear Dr. Shibuta,

Thank you for submitting your manuscript to PLOS ONE. After careful consideration, we feel that it has merit but does not fully meet PLOS ONE’s publication criteria as it currently stands. Therefore, we invite you to submit a revised version of the manuscript that addresses the points raised during the review process.

We look forward to receiving your revised manuscript.

Kind regards,

Dhermendra Tiwari

Academic Editor

PLOS ONE

Journal Requirements:

2. As part of your revision, please complete and submit a copy of the ARRIVE Guidelines checklist, a document that aims to improve experimental reporting and reproducibility of animal studies for purposes of post-publication data analysis and reproducibility: https://arriveguidelines.org/sites/arrive/files/Author%20Checklist%20-%20Full.pdf. Please include your completed checklist as a Supporting Information file. Note that if your paper is accepted for publication, this checklist will be published as part of your article.

Reviewers' comments:

Reviewer's Responses to Questions

**Comments to the Author**

1. Is the manuscript technically sound, and do the data support the conclusions?

Reviewer #1: Partly

Reviewer #2: Yes

Reviewer #3: No

2. Has the statistical analysis been performed appropriately and rigorously? 

Reviewer #1: Yes

Reviewer #2: Yes

Reviewer #3: Yes

3. Have the authors made all data underlying the findings in their manuscript fully available?

Reviewer #1: Yes

Reviewer #2: Yes

Reviewer #3: Yes

4. Is the manuscript presented in an intelligible fashion and written in standard English?

Reviewer #1: Yes

Reviewer #2: Yes

Reviewer #3: Yes

5. Review Comments to the Author

Reviewer #1: All abbreviations should be used full name in first using in abstract, introduction, and … .

Why was the PPF (PPF-PC) at 100 nM or 1 μM concentration diluted DMSO+PBS regimen chosen?

Why were the 17-day chosen to remove the fetuses?

What was the criteria for identifying a neuron?

The other techniques such as western blotting or RT-PCR should be used to confirm that cell death pathways (apoptosis, ..) is not involved in the administration of PPF.

The magnification and quality of fig. 4 and Fig. 5 low. The scale bar or magnification should be added to the figures.

What is the main difference between the present study and reference number 9?

Reviewer #2: Dear editor,

The study entitled “Effect of preconditioning on propofol-induced neurotoxicity during developmental period” presented a set of results showing that low concentrations of propofol using a preconditioning treatment did not alter the cytotoxic effects of propofol in moderate and high concentrations in fetal rat brain cells. The methodology used is satisfactory and in accordance with the conclusion presented. The manuscript is informative about the neurotoxicity shown by propofol during the embryogenesis of rats. After a minor review, the manuscript should be considered in PLOS ONE Journal.

Minor points:

In the introduction, please provide more robust data from literature where preconditioning treatments were positively functioning against a certain drug cytotoxicity.

Please provide the chemical structure of propofol.

Page 9, line 139: How many rat fetuses were used to obtain the 46 culture dishes of neurons?

Page 10, line 155: How many rat fetuses were used to obtain the 69 culture dishes of neurons?

Reviewer #3: The present study describes some interesting phrmacology of propofol in its neurotoxic capacity in the early development in experimental animals.

The works are interesting for further considerstion. However, befor the animal study, some basic and bridge translation between in vivo and in vitro are required. The [Ca2+] internalization via its specific chanels are not defined, although the event is extremely important. In addinio, its coupling with Na+ and K+ trsansporters are not explained through the experimental design. Other basic approaches they assessed are encouraged.

In conclusion, the study is interesting in our readers but the present form is too premature to justify the claims.

6. PLOS authors have the option to publish the peer review history of their article (what does this mean?). If published, this will include your full peer review and any attached files.

Reviewer #1: No

Reviewer #2: **Yes: **Leonardo Pereira Franchi

Reviewer #3: **Yes: **Cheorl-Ho Kim

---

## [Author Response · Author response to Decision Letter 0]

18 Jul 2022

Response to Reviewers

PONE-D-22-08060

Effect of preconditioning on propofol-induced neurotoxicity during the developmental period

PLOS ONE 

Dear Dr. Shibuta

Thank you for submitting your manuscript to PLOS ONE. After careful consideration, we feel that it has merit but does not fully meet PLOS ONE’s publication criteria as it currently stands. Therefore, we invite you to submit a revised version of the manuscript that addresses the points raised during the review process. 

If applicable, we recommend that you deposit your laboratory protocols in protocols.io to enhance the reproducibility of your results. Protocols.io assigns your protocol its own identifier (DOI) so that it can be cited independently in the future. For instructions see: https://journals.plos.org/plosone/s/submission-guidelines#loc- laboratory-protocols. Additionally, PLOS ONE offers an option for publishing peer-reviewed Lab Protocol articles, which describe protocols hosted on protocols.io. Read more information on sharing protocols at https://plos.org/protocols?utm_medium=editorial-email&utm_source=authorletters&utm_campaign=protocols. 

We look forward to receiving your revised manuscript. Kind regards, 

Dhermendra Tiwari Academic Editor PLOS ONE 

Journal Requirements:

2. As part of your revision, please complete and submit a copy of the ARRIVE Guidelines checklist, a document that aims to improve experimental reporting and reproducibility of animal studies for purposes of post-publication data analysis and reproducibility: https://arriveguidelines.org/sites/arrive/files/Author%20Checklist%20- %20Full.pdf. Please include your completed checklist as a Supporting Information file. Note that if your paper is accepted for publication, this checklist will be published as part of your article. 

Response

We submitted a copy of the ARRIVE guidelines with this letter. Thank you.

Upon re-submitting your revised manuscript, please upload your study’s minimal underlying data set as either Supporting Information files or to a stable, public repository and include the relevant URLs, DOIs, or accession numbers within your revised cover letter. For a list of acceptable repositories, please see http://journals.plos.org/ plosone/s/data-availability#loc-recommended-repositories. Any potentially identifying patient information must be fully anonymized. 

Response

We submitted data set as files “PPF 100n.xlsx”, “PPF-PCーSR-PO.xlsx” and “PC-PPF(imaging)-PO.xlsx”

Response

In accordance with your suggestion, we deleted “data not shown” and revised. 

Reviewers' comments:

Reviewer's Responses to Questions

Comments to the Author 

1. Is the manuscript technically sound, and do the data support the conclusions? 

Reviewer #1: Partly Reviewer #2: Yes Reviewer #3: No 

2. Has the statistical analysis been performed appropriately and rigorously? 

Reviewer #1: Yes

Reviewer #2: Yes

Reviewer #3: Yes 

3. Have the authors made all data underlying the findings in their manuscript fully available? 

Reviewer #1: Yes Reviewer #2: Yes Reviewer #3: Yes 

4. Is the manuscript presented in an intelligible fashion and written in standard English? 

Reviewer #1: Yes Reviewer #2: Yes Reviewer #3: Yes 

5. Review Comments to the Author 

Response to Reviewer #1: 

Thank you very much for your valuable comments. Your comments have helped us improve our manuscript. We have stated the changes made to the manuscript and have answered your questions below. I hope these revisions meet your approval.

All abbreviations should be used full name in first using in abstract, introduction, and ... . 

Response

We confirmed that all abbreviations were spelled out at their first occurrence, as per your comments.

Why was the PPF (PPF-PC) at 100 nM or 1 μM concentration diluted DMSO+PBS regimen chosen? 

Response

PPF at 1 μM did not kill neurons significantly in our preliminary experiment. This concentration was the maximum concentration that did not affect the survival rate of the primary cultured neurons. However, 10 μM PPF killed neurons significantly. We also used PPF at 100 nM in case PPF 1μM affected neuronal survival (but it did not) or PPF-PC improved survival rate (but it did not).

Why were the 17-day chosen to remove the fetuses?

Response

Cultures of the embryonic and fetal mammalian central nervous system have been used in many experiments due to its reliability, while the in vitro maintenance of adult mammal neurons has hitherto been largely unsuccessful. We identified that E16–E17 was the best survival rate until 28DIV. The brains of fetuses on E<15 were too small to remove, and therefore, sometimes, we could not collect as many neurons as we anticipated.

What was the criteria for identifying a neuron? 

Response

As we previously described, in order to confirm the purity of the neuronal culture, cells were immunostained with anti-MAP2 or anti-GFAP antibody, before and after the experiment. More than 97% expressed MAP2 and less than 2% of the cells expressed GFAP, regardless of the duration of the experiments. This demonstrated that most of the cells in our cultures were neurons. (J Neurosci Res. 72(5):613-21, 2003; Br. J. Anaesth. 104: 52-58, 2010)

The other techniques such as western blotting or RT-PCR should be used to confirm that cell death pathways (apoptosis, ..) is not involved in the administration of PPF.

Response

Moderate to high concentrations of PPF administration during the developmental period (till DIV8) are involved in neuronal death, as shown in our previous reports (Neurotoxicology 69: 320-9, 2018). In the present study, our main purpose was to investigate whether PC alleviates PPF-induced neurotoxicity but not to elucidate cell death pathways. To minimize the number of experimental animals we used, we did not investigate death pathways in the present study.

The magnification and quality of fig. 4 and Fig. 5 low. The scale bar or magnification should be added to the figures. 

Response

We added scale bars in the Figures following your valuable comments. 

What is the main difference between the present study and reference number 9? 

Response

Reference 9 (our previous report, 2018) demonstrated that exposure to moderate – high doses of PPF during the early developmental period leads to neural death. Based on this finding, we tried preconditioning treatments to alleviate this PPF-induced neurotoxicity. PC with low-dose NMDA was effective against subsequent glutamate insults, as Sragovich et al (2012) demonstrated. However, our result showed that PPF-PC did not improve survival rate of the neurons from later moderate – high doses of PPF exposure during the early developmental period.

Thank you very much for your valuable comments, again.

Reviewer #2: Dear editor,

The study entitled “Effect of preconditioning on propofol-induced neurotoxicity during developmental period” presented a set of results showing that low concentrations of propofol using a preconditioning treatment did not alter the cytotoxic effects of propofol in moderate and high concentrations in fetal rat brain cells. The methodology used is satisfactory and in accordance with the conclusion presented. The manuscript is informative about the neurotoxicity shown by propofol during the embryogenesis of rats. After a minor review, the manuscript should be considered in PLOS ONE Journal. 

Response to reviewer #2

Thank you for your valuable comments. Your comments encouraged us to improve our manuscript.

We have stated the changes made to the manuscript and have answered your questions below. I hope these revisions meet your approval.

Minor points:

In the introduction, please provide more robust data from literature where preconditioning treatments were positively functioning against a certain drug cytotoxicity.

Response

According to your helpful suggestion, we have added two references (Sragovich, S. et al 2012 and Navon et al 2012). The addition of these two references helped improve our manuscript significantly.

Please provide the chemical structure of propofol.

Response

According to your comment, we have provided the chemical structure of PPF as Figure 1.

Page 9, line 139: How many rat fetuses were used to obtain the 46 culture dishes of neurons?

Page 10, line 155: How many rat fetuses were used to obtain the 69 culture dishes of neurons? 

Response

I am unable to provide the exact number of rat fetuses. However, we usually obtain approximately 20–30 cultured dishes from one pregnant rat. Each pregnant rat has 4–12 fetuses. In the present study, we confirmed that we purchased and used 12 pregnant rats in total. We mentioned this in the materials and method section of the revised manuscript.

Reviewer #3: The present study describes some interesting phrmacology of propofol in its neurotoxic capacity in the early development in experimental animals.

The works are interesting for further considerstion. However, befor the animal study, some basic and bridge translation between in vivo and in vitro are required. The [Ca2+] internalization via its specific chanels are not defined, although the event is extremely important. In addinio, its coupling with Na+ and K+ trsansporters are not explained through the experimental design. Other basic approaches they assessed are encouraged. 

In conclusion, the study is interesting in our readers but the present form is too premature to justify the claims. 

Response

Thank you very much for your valuable comments. 

The involvement of L-type calcium channels in PPF-induced neurotoxicity has already been reported by Kahraman et al (2008). Our preliminary studies also showed the same result, therefore, we did not perform further investigations on this. We understand that a major limitation of this experiment is that the degree of effect on cortical neurons in vitro may not necessarily correlate with the neurotoxic effect in vivo. In addition, we comprehend the importance of the involvement of Na+ and K+ transporters in PPF-induced neurotoxicity to clarify the neurotoxic mechanism as you mentioned. We are greatly inspired by your valuable comments for our future study. Still, we believe the alleviation of neurotoxicity is one of the main interests of not only anesthesiologists, pediatricians, and obstetricians, but all medical doctors. Since preconditioning is considered to be one of the useful therapies, in addition to hypothermia and medication as a neuroprotection tool, we believe PlosOne will provide the appropriate audience for our research.

6. PLOS authors have the option to publish the peer review history of their article. If published, this will include your full peer review and any attached files. 

Do you want your identity to be public for this peer review? For information about this choice, including 

consent withdrawal, please see our Privacy Policy. Reviewer #1: No

Reviewer #2: Yes: Leonardo Pereira Franchi Reviewer #3: Yes: Cheorl-Ho Kim 

---

## [Decision Letter · Decision Letter 1]

5 Aug 2022

Effect of preconditioning on propofol-induced neurotoxicity during the developmental period

PONE-D-22-08060R1

Dear Dr. Shibuta,

We’re pleased to inform you that your manuscript has been judged scientifically suitable for publication and will be formally accepted for publication once it meets all outstanding technical requirements.

Kind regards,

Dhermendra Tiwari

Academic Editor

PLOS ONE

Additional Editor Comments (optional):

Reviewers' comments:

Reviewer's Responses to Questions

**Comments to the Author**

1. If the authors have adequately addressed your comments raised in a previous round of review and you feel that this manuscript is now acceptable for publication, you may indicate that here to bypass the “Comments to the Author” section, enter your conflict of interest statement in the “Confidential to Editor” section, and submit your "Accept" recommendation.

Reviewer #3: (No Response)

2. Is the manuscript technically sound, and do the data support the conclusions?

Reviewer #3: Yes

3. Has the statistical analysis been performed appropriately and rigorously? 

Reviewer #3: Yes

4. Have the authors made all data underlying the findings in their manuscript fully available?

Reviewer #3: Yes

5. Is the manuscript presented in an intelligible fashion and written in standard English?

Reviewer #3: Yes

6. Review Comments to the Author

Reviewer #3: The authors revised the previous manuscript, cordingly,. No more questions are raised by the present reviewer.

The findings are usable for the specific field researchers or clinicians, as the drugs are routinly used in the clinics

7. PLOS authors have the option to publish the peer review history of their article (what does this mean?). If published, this will include your full peer review and any attached files.

Reviewer #3: No

---

## [Editor Report · Acceptance letter]

11 Aug 2022

PONE-D-22-08060R1 

Effect of preconditioning on propofol-induced neurotoxicity during the developmental period 

Dear Dr. Shibuta:

I'm pleased to inform you that your manuscript has been deemed suitable for publication in PLOS ONE. Congratulations! Your manuscript is now with our production department. 

Kind regards, 

on behalf of

Dr. Dhermendra Tiwari 

Academic Editor

PLOS ONE